# OpenReview forum: "MPO: An Efficient Post-Processing Framework for Mixing Diverse Preference Alignment"
_ICML.cc/2025/Conference — ICML 2025 poster_

### Official Review · Reviewer_MZtJ · 2025-02-25

**Overall Recommendation:** 3

**Summary:**

This paper proposes MPO, an efficient post-processing framework for mixing diverse preference alignment. The authors use batch stochastic mirror descent to find the optimal combination coefficients for output combination.


## update after rebuttal

Most of the concerns are resolved, so the reviewer raises the score to weak accept.

**Claims And Evidence:**

Yes.

**Essential References Not Discussed:**

None.

**Experimental Designs Or Analyses:**

Yes.

**Methods And Evaluation Criteria:**

Yes.

**Other Comments Or Suggestions:**

None.

**Other Strengths And Weaknesses:**

- Strengths

    - The proposed method is training-free. Theoretically, the authors can combine several existing well-aligned LLMs to get the desired LLM without any cost training process.
    - The authors provide theoretical guarantees for the proposed method.


- Weaknesses

    - Lack of comprehensive evaluation of the proposed method.
    - The reviewer doubts the feasibility of serving multiple LLMs simultaneously in practice.

**Questions For Authors:**

- According to Algorithm 1, do the authors simply combine the output logits of different LLMs? From Figure 1, the reviewer thinks the authors conduct something like model merging.

- The reviewer thinks the experimental evaluation of the method is not enough.

    - In Table 1, do the authors adopt $\pi_{helpful/harmless/humorous}$ provided by others or train them by yourselves? If the authors train these models by yourselves, the reviewer thinks the authors should add comparison results with models provided by others. For example, comparison with PKU-SafeRLHF [1] in terms of helpfulness and harmlessness.

    - If we consider the win rate of the reference model as 50%, the aligned model is only marginally better than the baseline. Are such results reasonable?
    - Following the above point, in Table 1, a large $\beta$ leads to better results, this is quite uncommon because general alignment algorithms like PPO usually use a small constraint. Can we think a large $beta$ represents a similar model as the original one? If so, what is the role of the alignment method during the process?

- In practice, the reviewer thinks it is quite difficult for us to serve multiple LLMs at the same time. Especially when the model is large. If the authors adopt the model merging method, this should not be a problem.

[1] https://huggingface.co/datasets/PKU-Alignment/PKU-SafeRLHF

**Relation To Broader Scientific Literature:**

This paper proposes a post-processing method for diverse preference optimization. From the reviewer's perspective, it is novel compared to previous scientific literature.

**Theoretical Claims:**

The reviewer assumes all proofs are correct.

---

> ### Author Rebuttal · Authors · 2025-04-01
>
> We greatly appreciate your constructive and insightful feedback! Here we provide a detailed response to address all of your concerns below.
>
> > Confusion of Algorithm 1 and Figure 1.
>
> Thank you for the question and apologies for any confusion. As shown in Thm3.4, we have
> $$\pi^*(y|x) \propto \prod_{k=1}^K\left(\pi_k(y|x)\right)^{\lambda_k^*},$$
> which can be seen as a logit-level aggregation of multiple LLMs. However, our main goal is to compute $\lambda^*$ that maximizes the minimum reward, which balances the trade-offs among different objectives. Based on your comment, we will revise Figure 1 to better highlight the role of $\lambda$ in shaping the final policy.
>
> > Evaluation of $\pi_{helpful/harmless/humorous}$ in Table 1.
>
> Thank you for your feedback.
> 1. $\pi_{helpful/harmless/humorous}$ in Table 1 were trained by our own, and the results demonstrate a significant surplus in the corresponding rewards—indicating the effectiveness of our single-objective policies.
> 2. However, our primary goal is to balance multiple objectives rather than to optimize individual objectives. Given the differences in training data, we believe that direct comparisons of single-objective policies may not be entirely fair.
> 3. Nonetheless, we have added further comparisons with the PKU-SafeRLHF model
> with MPO under $\beta=0.1$ in Eq.10. As shown in FIgure 4, our $\pi_{MPO}$ mostly rely on $\pi_{helpful}$ and $\pi_{harmless}$, which aligns closely with the objectives considered in the PKU-SafeRLHF. The results below show that MPO still achieves the highest minimum win rate.
>
> Table : Win rate(%) against the Reference Model
> | Model  | Helpful  | Harmless  | Humorous   | Min |
> |-|:-:|:-:|:-:|:-:|
> | $\pi_{PKU}$ | 53.1 | 40.8 | 56.1 | 40.8 |
> | $\pi_{MPO}$ 	| 46.3 | 53.1 | 54.1 | $\color{red}{46.3}$ |
>
> 4. Moreover, the normalized reward for $\pi_{MPO}$ and $\pi_{PKU}$ are listed below:
>
> |Model | $r_{helpful}$  | $r_{harmless}$  | $r_{humorous}$ |
> |-:|:-:|:-:|:-:|
> | $\pi_{MPO}$ | -0.176 | 0.564 | 0.104 |
> | $\pi_{PKU}$ | 0.150  | -0.05  | 0.150 |
>
> Here, larger normalized rewards indicate better alignment with the corresponding objective.
> These results indicate that the PKU-SafeRLHF model places a greater emphasis on helpfulness, which is likely due to its constrained optimization loss. We will add more discussion with PKU-SafeRLHF in the revision.
>
> > Aligned model is only marginally better than the baseline. Are such results reasonable?
>
> Thank you for the question.
> 1. Table 1 demonstrates MPO's optimality among all methods under the max-min setting, with the aligned model outperforming the baseline on all individual objectives.
> 2. The improvement appears only marginal because of the inherent conflict between objectives such as helpfulness and harmlessness. Multi-objective alignment tasks involve complex trade-offs, and simultaneously improving all conflicting objectives is nearly impossible [1]. This challenge is precisely why we consider the max-min setting in our work.
>
> References:
> [1] Rame, A, et al. Rewarded soups: towards pareto-optimal alignment by interpolating weights fine-tuned on diverse rewards. NeurIPS 2023.
>
> > What is the role of the $\beta$ during the alignment process?
>
> Thank you for your question.
> 1. While a larger $\beta$ encourages policy to stay close to the reference model, it plays a critical role in balancing conflicting objectives in our multi-objective setting. Stronger regularization helps stabilize aggregation and maintain desirable baseline behaviors while optimizing the minimum reward.
> 2. $\beta$ is a tunable hyperparameter. As shown in Table 1, $\beta=0.5$ outperforms $\beta=0.1$ and $\beta=\infty$. In particular, the case of $\beta=\infty$ collapsed to the reference policy with a 50% win rate. This highlights the importance of tuning $\beta$ for optimal performance.
>
> > In practice, the reviewer thinks it is quite difficult for us to serve multiple LLMs at the same time. Especially when the model is large. If the authors adopt the model merging method, this should not be a problem.
>
> 1. We understand the concern about serving multiple LLMs concurrently. However, the efficiency of MPO primarily comes from the training phase. In Section 4.2 of our experiments, training policies using PPO-based approaches (e.g., MaxMin-RLHF, MORLHF) requires approximately 10 A100 GPU hours, whereas MPO only requires around 2.5 A100 GPU hours, since it avoids reinforcement learning step.
> 2. In the inference phase, instead of running full inference on all LLMs simultaneously, we can compute single-objective policy outputs logits in parallel, then aggregate logits using the MPO framework.
> 3. Additionally, as shown in Table 1, our method achieves better alignment than parameter merging baselines like reward soups and future work will focus on further optimizing the inference pipelines to enhance the scalability of MPO in real-world applications.

---

### Official Review · Reviewer_FxzW · 2025-03-12

**Overall Recommendation:** 2

**Summary:**

This paper studies how to align diverse objectives of human preferences.  The authors propose a post-processing approach that combines the optimal policies for each objective without requiring retraining from scratch. Moreover, the author also studies the max-min RLHF, and shows that we can find the optimal policy by adjusting the weight using mirror descent. Experimental results are provided to support their theoretical findings.

**Claims And Evidence:**

Yes. The claims are correct, clear and easy to follow.

**Essential References Not Discussed:**

Two essential references that are not discussed are (Shi et al., 2024) and (Ramesh et al., 2024). The former presents a result similar to Theorem 3.4 in this paper, while the latter introduces a similar idea of adjusting weights using mirror descent to achieve max-min goal.

**Experimental Designs Or Analyses:**

The experiment compares MPO with previous algorithms such as Reward Soups and Max-Min RLHF, along with baselines like the reference model and single-reward algorithm. My main concern is that the authors should include comparisons with more prior works, such as MOD (Shi et al., 2024), as well as a baseline algorithm that aggregates the reward and trains the model directly on it. Additionally, since Algorithm 1 has learned the optimal weight $\lambda$, the authors could utilize this weight to implement RS instead of using a uniform weight $[1/3,1/3,1/3]$.

Shi R, Chen Y, Hu Y, et al. Decoding-time language model alignment with multiple objectives. NeurIPS 2024.

**Methods And Evaluation Criteria:**

The authors evaluate their algorithms using a classical MORLHF dataset named HH-RLHF, which contains three objectives Helpful, Harmless, and Humorous.

**Other Comments Or Suggestions:**

The author should provide more theoretical comparisons (and if possible, empirical comparisons) between MPO and previous algorithms, and the novelty of MPO.

**Other Strengths And Weaknesses:**

The strengths and weaknesses are provided above.

**Questions For Authors:**

1. In Line 139, the author states that "Balancing multiple, often competing objectives leads to training instability, while the need to train multiple reward models and perform RL updates makes them computationally expensive." However, obtaining the optimal policy for each objective also requires training multiple reward models. Therefore, I do not think that the computational cost of MPO is lower than that of previous algorithms like Reward Soups. Could the authors provide further clarification on why MPO is not computationally expensive?

**Relation To Broader Scientific Literature:**

The key contribution of this paper is the proposed algorithm that enables language models to align with diverse objectives by leveraging the optimal policy for each objective, rather than requiring training from scratch. However, the novelty appears to be limited, as the main theorem (Theorem 3.4) has already been studied in Theorem 1 of (Shi et al., 2024). Could the authors clarify the differences between their main theorem and the one in (Shi et al., 2024)? Additionally, the use of mirror descent for weight adjustment closely resembles the approach in (Ramesh et al., 2024). As a result, it remains unclear whether this paper offers a novel theoretical contribution.


Shi R, Chen Y, Hu Y, et al. Decoding-time language model alignment with multiple objectives. NeurIPS 2024.

Ramesh S S, Hu Y, Chaimalas I, et al. Group robust preference optimization in reward-free rlhf. NeurIPS 2024.

**Theoretical Claims:**

I check the proof and it is correct to me.

---

> ### Author Rebuttal · Authors · 2025-04-01
>
> We greatly appreciate your constructive and insightful feedback! Here we provide a detailed response to address all of your concerns.
>
> > Differences between main theorem and the one in Shi et al., 2024
>
> Thank you for your feedback. Compared to Thm 1 in Shi et al., our approach differs in both objective and applicable setting, with some overlap in a special case:
> 1. Our Thm 3.4 addresses the max-min setting without explicit preference weights, whereas Shi et al. uses predefined weights. In the special case of linear reward aggregation, Lem 3.9 shows that the optimization leads to a closed-form solution, which indeed reaches same conclusion in Shi et al.
> 2. We introduce an auxiliary normalizing operator for rewards, which is crucial for transforming the optimization over $\lambda$ into Eq12 (line 199). Without it, the reward function cannot be avoided in optimization.
> 3. In terms of the derivation, Shi et al. uses a Legendre transform, converting the problem into
> $$\max_y \pi_{ref}(y|x) \text{~~such that~} r(y∣x)>C,$$
> where $C$ is unspecified. In contrast, our proof uses a direct reward–policy mapping, leading to the closed-form expression for $\pi^*$ directly, providing a more interpretable and transparent theoretical derivation.
>
> > The use of mirror descent for weight adjustment closely resembles the approach in Ramesh et al., 2024.
>
> Indeed the high-level idea of Ramesh et al. is to perform robust alignment, which is similar to us. However, we highlight several differences in terms of less resources required, applicable settings, and methodology developed.
> 1. Ramesh et al. aims to derive a group robust preference optimization objective and conduct robust alignment from scratch, which is computationally expensive and requires extensive hyperparameter tuning. In contrast, we could directly use existing single-objective policies, avoiding full retraining and significantly reducing computational cost.
> 2. By reusing pretrained or open-source LLMs, we simplify robust training to only updating preference weights—a lightweight post-processing step that aligns better with practical academic and industry use cases. Ramesh et al. 's method is better suited for settings where LLMs must be trained from scratch.
> 3. Methodologically, Ramesh et al. uses a gradient descent-mirror ascent method to update the policy and the weight simultaneously as its objective is to solve a min-max optimization. In addition, it has access to unbiased gradient estimators. In our case, since we do not train policies, weight optimization becomes a conditional stochastic problem without unbiased  gradient estimators. To tackle this, we designed a biased mirror descent method and analyzed its convergence.
>
> > Could the authors provide further clarification on why MPO is not computationally expensive?
>
> Thanks for your question.
> 1. While standard RLHF requires training multiple reward models, this can be avoided by using DPO, which is mathematically equivalent and was adopted in our experiments for efficiency.
> 2. For $dim(\lambda=3)$, training the policy via MORLHF (or maxmin-RLHF) for a fixed $\lambda$ takes approximately 10 A100 GPU hours, whereas solving for $\lambda$ in MPO only takes 2.5 A100 GPU hours.
> 3. Unlike standard MORLHF,  which re-runs PPO for different $\lambda$, MPO avoids this overhead by efficiently combining logits. While Reward Soups has similar cost with predefined weights, MPO achieves better alignment (shown Table 1), highlighting its effectiveness in balancing multiple onjectives.
>
> > Comparisons with more prior works.
>
> Thank you for the suggestion, we have added further comparisons with prior works:
> 1. We compare our approach with MORLHF, which employs linear aggregated rewards using PPO, as well as with reward soups that utilize learned weights. As shown in Table 1, our MPO method still achieves the highest minimum win rate, where $\pi_{Weighted RS}$
> corresponds to the Reward Soups with learned weights
>
> | Model  | Helpful  | Harmless  | Humorous   | Min |
> |-|:-:|:-:|:-:|:-:|
> |||$\beta = 0.1$|
> | $\pi_{RS}$ 		| 44.8 | 59.4 | 56.4 | 44.8 |
> | $\pi_{Weighted RS}$ | 45.4 | 52.2 | 51.3 | 45.4 |
> | $\pi_{MORLHF}$ 	| 42.9 | 56.7 | 54.5 | 42.9 |
> | $\pi_{MPO}$ 		| 46.3 | 53.1 | 54.1 | $\color{red}{46.3}$ |
> |||$\beta = 0.5$|
> | $\pi_{RS}$ 		| 51.9 | 53.7 | 50.0 | 50.0 |
> | $\pi_{Weighted RS}$ | 53.7 | 50.8 | 48.8 | 48.8 |
> | $\pi_{MORLHF}$ 	| 41.7 | 54.4 | 52.9 | 41.7 |
> | $\pi_{MPO}$ 		| 54.9 | 53.1 | 57.1 | $\color{red}{53.1}$ |
> Table 1: Win rate(%) against the Reference Model
>
> 2. While MOD employs a linear combination of logits, in Figure 2(b), we have evaluated
> $$\pi(y|x) \propto \pi_1(y|x)^{\lambda_1}\pi_2(y|x)^{1-\lambda_1}$$
> with different $\lambda_1 \in \Lambda_{grid}= \\{0.0, 0.2, 0.4, 0.6, 0.8, 1.0\\}$. Our results show that the policy obtained via MPO achieves the best objective performance across this grid, outperforming MOD with predefined weights in the max-min setting.

---

> > ### Comment · Reviewer_FxzW · 2025-04-07
> >
> > Thanks for the response. However, I still think this paper is very similar to the main theorem in [Shi et al. 2024], with a similar weight adjustment in [Ramesh et al. 2024]. Hence, the novelty seems limited. I will keep my score.

---

> > > ### Author Response · Authors · 2025-04-07
> > >
> > > Dear Reviewer FxzW,
> > >
> > > Thank you for your follow-up comment. We appreciate your continued engagement with our work. However, we respectfully disagree with your assessment and would like to offer a clarification of the key differences between our approach and the references you mentioned.
> > >
> > > First, the primary focus of our paper is on multi-objective alignment via a max-min formulation that does not require pre-specified preference weights. This focus is *very different* from that of [Shi et al. 2024], which assumes fixed weights as input. Yet how to choose such a fixed weight can be highly nontrivial in practice. On the contrary, our method is specifically designed to infer a robust solution without relying on explicit prior knowledge of user preferences.
> > >
> > > Second, as noted in our rebuttal, we introduce an *auxiliary normalization operator* on the reward functions, which plays a critical role in enabling a closed-form expression for the optimization over $\lambda$ (see Eq. 12). This normalization step changes the analytical landscape and requires a different theoretical treatment than that in [Shi et al. 2024].
> > >
> > > Regarding your comment on "a similar weight adjustment" in [Ramesh et al., 2024], we would like to emphasize 1) from a computational perspective, our method can directly use existing policies, avoid policy retraining, and thus is much cheaper; 2) from a technical point of view, to achieve such a computational reduction, we propose to use a novel  *biased mirror descent* to update the weights, which makes it possible to keep the policy fixed. This is fundamentally different from the approach in [Ramesh et al. 2024], which employs a joint min-max optimization framework and thus involves both updating weights and policy retraining at each step. In particular, they have access to unbiased gradient estimators of the weight at the costs of much higher computational costs. That being said, our biased optimization method is tailored to our setting to avoid updating the policy and ensure smaller computational costs. In addition, we provide a novel convergence analysis.

---

### Official Review · Reviewer_sv4s · 2025-03-15

**Overall Recommendation:** 2

**Summary:**

This work proposes Mixing Preference Optimization (MPO), a post-processing framework for aggregating single objective policies with a mixing of preference alignment. Specifically, the authors combine two multi-objective RLHF approaches, MORLHF and MaxMin-RLHF using a post-processing strategy to combine single-objective policies. The overall objective of MPO can be viewed as a max-min game between the policy and the combinational factor $\lambda$ of MORLHF.


**After rebuttal:**
Thank you to the authors for their feedback. However, my concerns regarding [Theoretical Claims] remain unresolved. In particular, the primary issue is that the correctness of Equation (10) has not been substantiated. This affects the overall confidence in the correctness of the proposed method. I will maintain my original score.

**Claims And Evidence:**

The authors claim that  MPO significantly reduces training costs and computational overhead. However, I can not find any theoretical or empirical analysis of the computational complexity compared with the baselines such as MORLHF and MaxMin-RLHF.

**Essential References Not Discussed:**

Not found.

**Experimental Designs Or Analyses:**

I have some concerns about the experimental design and analyses as follows.

1. The size of $\lambda$ is so small (2 or 3) in the experiments. This can make the soundness of the proposed algorithm weak in practical usage.

2. The comparison does not consider other important RLHF baselines such as PPO-based approaches. The existing work has proposed to aggregate the reward model of PPO to mix diverse preference alignment.

3. The analysis of training costs is missing in the experimental analyses. Besides, it is unclear whether the proposed method will improve the computational complexity in the inference phase.

**Methods And Evaluation Criteria:**

The proposed method uses a post-processing algorithm to tackle alignment with diverse human preferences. The method, itself, is easy to follow and the evaluation criteria make sense.

**Other Comments Or Suggestions:**

N/A

**Other Strengths And Weaknesses:**

For strengths, 1. the proposed method is easy to follow for mixing diverse preference alignment. 2. The problem of alignment with diverse human preferences is novel and interesting within the area of RLHF.

For weaknesses, 1. the writing is needed to improve, e.g., section 3 is hard to follow. 2. the contribution and novelty are marginal to some extent, the proposed algorithm is a straightforward combination of two previous works MORLHF and MaxMin-RLHF.

**Questions For Authors:**

1. How can the correctness of Eq. 10 be verified?

2. Whether the proposed method will improve the computational complexity in the inference phase?

3. Can we improve the size of $\lambda$ in the experiments?

4. Where are the key novelty and soundness compared with MORLHF and MaxMin-RLHF?

**Relation To Broader Scientific Literature:**

An improvement method on the alignment of diverse human preferences upon previous work such as MORLHF and MaxMin-RLHF.

**Theoretical Claims:**

The main concerns are the correctness of the derivation of the algorithm, specifically:

1. The correctness of Eq. 10 can not be verified, which is an important premise for deriving the optimization objective in Theorem 3.4. (Main Theorem).

2. $\pi^*$ is a global and certain optimality of the policy, but $r^*$ is obtained by stochastic descent in Alg. 2. It is unclear how the estimation of  $r^*$ can approximate optimize the objective of Eq. 10.

3. The optimization of $\pi^*$ is with respective to the input-output $(x,y)$. However, the optimization of $r^*$ is input-output agnostic. This anti-intuitive result makes me concerned about the soundness of the proposed method in comparison with MaxMin-RLHF.

---

> ### Author Rebuttal · Authors · 2025-04-01
>
> We greatly appreciate your constructive and insightful feedback! Here we provide a detailed response to address all of your concerns.
>
> > How can the correctness of Eq. 10 be verified?
>
> Thanks for the question. As we explained after Eq. 10 in our original submission, the result follows directly from Sion’s minimax theorem. The objective function is convex in $\lambda$ ($\pi_{\theta}$ fixed)​ and concave in $\pi_{\theta}$​ ($\lambda$ fixed), which satisfies the conditions for the theorem and guarantees the interchange of max and min in Eq. 10.
>
> > $\pi^*$  is a global and certain optimality of the policy, but $r^*$ is obtained by stochastic descent in Alg. 2. It is unclear how the estimation of $r^*$ can approximate optimize the objective of Eq. 10.
>
> 1. If $r^*$ refers to the reward model, our approach does not involve reward training; Instead, we use an auxiliary normalizing operator to directly derive the optimal policy
> $$\pi^*(y|x) \propto \prod_{k=1}^K\left(\pi_k(y|x)\right)^{\lambda_k^*}$$
> 2. If $r^*$ refers to the preference weight $\hat{\lambda}$ as solved by Alg. 2, then Thm 3.8 provides a KL-based error bound between the learned policy $\hat{\pi}$ and $\pi^*$. The theorem formally states that $\hat{\pi}$ closely approximates $\pi^*$ under mild conditions.
>
> > The optimization of $\pi^*$ is with respect to the input-output (x,y). However, the optimization of $r^*$ is input-output agnostic.
>
> Thank you for your question. Again, we believe you are referring to $\lambda$. As shown in Thm 3.4, our main task is to solve for $\lambda^*$ via Eq .12. Once obtained, the optimal policy is effectively a linear combination of the logits from the single-objective policies. Additionally, when applying Alg 2, we only need the individual policies and a set of prompts.
>
> > Can we improve the size of $\lambda$ in the experiments?
>
> Thank you for your comment. We would like to point out that $dim⁡(\lambda)=3$ is consistent with prior works [1,2,3], which also adopts up to three objectives. This dimensionality has proven effective for capturing the scalability of multi-objective alignment, balancing soundness and computational efficiency.
>
> References:
> 1. Chakraborty, S, et al. MaxMin-RLHF: Alignment with diverse human preferences. ICML 2024.
> 2. Yang, R, et al. Rewards-in-context: Multi-objective alignment of foundation models with dynamic preference adjustment. NeurIPS 2024.
> 3. Zhou, Z, et al. Beyond one-preference-fits-all alignment: Multi-objective direct preference optimization. arXiv preprint arXiv:2310.03708 (2023).
>
> > The comparison does not consider other important RLHF baselines such as PPO-based approaches.
>
> Thank you for the comment. Standard MORLHF approaches [1,2] optimize linearly aggregated reward functions. while Maxmin-RLHF perform PPO to obtain the policy by optimizing:
> $$\max_{\pi} \min_k E\left[r_{k}(x,y) \right] - \beta D_{\mathrm{KL}}\left[\pi \Vert \pi_{\text{ref}}\right].$$
> In light of your comments, we have added more comparisons of such methods. Notably, our MPO method still achieves the highest minimum win rate.
>
> | Model  | Helpful  | Harmless  | Humorous   | Min |
> |-|:-:|:-:|:-:|:-:|
> |||$\beta = 0.1$|
> | $\pi_{Maxmin-RLHF}$	| 44.6 | 56.1 | 51.4 | 44.6 |
> | $\pi_{MORLHF}$ 	| 42.9 | 56.7 | 54.5 | 42.9 |
> | $\pi_{MPO}$ 		| 46.3 | 53.1 | 54.1 | $\color{red}{46.3}$ |
> |||$\beta = 0.5$|
> |  $\pi_{Maxmin-RLHF}$	| 46.1 | 53.8 | 54.8 | 46.1 |
> | $\pi_{MORLHF}$ 	| 41.7 | 54.4 | 52.9 | 41.7 |
> | $\pi_{MPO}$ 		| 54.9 | 53.1 | 57.1 | $\color{red}{53.1}$ |
>
> References:
> 1. Ji, J, et al. Beavertails: Towards improved safety alignment of llm via a human-preference dataset. NeurIPS 2023.
> 2. Wu, Z, et al. Fine-grained human feedback gives better rewards for language model training. NeurIPS 2023.
>
> > The analysis of training costs is missing in the experimental analyses.
>
> Thanks for the suggestion. For computation cost, training the policy using MORLHF (or maxmin-RLHF) with aggregated reward models takes approximately 10 A100 GPU hours, since both methods rely on PPO for policy optimization and differ only in how they aggregate reward functions,  In contrast, our approach avoids PPO entirely, solving for the preference weights $\lambda$ via Alg 2 requires only about 2.5 A100 GPU hours, offering a significant reduction in training time while still achieving competitive performance.
>
>
> > Where are the key novelty and soundness compared with MORLHF and MaxMin-RLHF?
>
> Thanks for the comment, We have stated the key novelty and soundness of MPO compared to MORLHF and MaxMin-RLHF in both the Introduction and Conclusion sections. In summary:
> 1. MPO connects reward aggregation to policy aggregation, yielding a closed-form aggregation rule. 2. Our approach operates directly on single-objective policies, avoiding extra RL updates and reward model training, thereby significantly reducing computational overhead. 3. The method is backed by rigorous theoretical error bounds that ensure robustness relative to the optimal policy.

---

### Official Review · Reviewer_e4hE · 2025-03-17

**Overall Recommendation:** 4

**Summary:**

This paper proposes MPO, a  framework designed to mix diverse single-objective policies for aligning LLMs with human preferences. Instead of training a costly multi-objective RLHF model from scratch, this paper shows how pre-trained, single-objective policies can be aggregated using a batch stochastic mirror descent (BSMD) algorithm. This paper derives a closed-form solution relating the aggregated policy to the individual policies and provide theoretical guarantees. Experiments on multiple multi-objective preferences tasks illustrate that MPO can outperform baseline methods while having lower computation cost.

**Claims And Evidence:**

Yes, most of the claims are supported by the theoretical guarantees and experiment results.

**Essential References Not Discussed:**

This paper discusses a wide range of relevant works of RLHF, preference learning and the diverse alignment objectives.

**Experimental Designs Or Analyses:**

The experimental design is sound in that it considers multiple objectives and compares MPO with representative baselines.  One potential issue is that the evaluation relies on GPT 3.5/4 models, although common in current research, may be sensitive to prompt design.

**Methods And Evaluation Criteria:**

The evaluation mainly relies on the judgment of GPT 3.5/4, it would be better to include human validation.

**Other Comments Or Suggestions:**

Please see the weakness.

**Other Strengths And Weaknesses:**

**Strengths**

1. This paper proposes an effective method MPO that utilizes existing single-objective policy to combine into a unified one for diverse preferences alignment. The proposed method is well formalized and supported by theoretical analysis.
2. MPO has a clear advantage in efficiency, which avoids alignment from scratch.
3. Experiment results demonstrate across a wide range of alignment tasks demonstrate the effectiveness of MPO.

**Weaknesses**

1. It would be better to add more validations on the GPT-based evaluation.
2. It would be better to include a discussion on the scalability of MPO with respect to computation resources when increasing the number of objectives.

**Questions For Authors:**

Please see the weakness.

**Relation To Broader Scientific Literature:**

This paper makes meaningful contribution to the preference learning and alignment of LLMs.

**Theoretical Claims:**

Yes, I carefully checked the key steps of the aggregated policy and convergence guarantees for BSMD, and they appear to be correct.

---

> ### Author Rebuttal · Authors · 2025-04-01
>
> We greatly appreciate your constructive and insightful feedback! Here we provide a detailed response to address all of your concerns below.
>
> > It would be better to add more validations on the GPT-based evaluation.
>
> Thanks for the suggestion. When utilizing GPT-based evaluations, we have experimented with multiple GPT versions and leveraged prompts similar to previous works such as [1] and [2]. We found that while there are some variations in the output, the overall performance trends remain stable, which gives us confidence in the reliability of this evaluation method. We are also open to further validations and comparisons to ensure that our evaluation framework is as comprehensive and robust as possible.
>
> References:
> 1. Rafael Rafailov, Archit Sharma, Eric Mitchell, Christopher D Manning, Stefano Ermon, and Chelsea Finn. Direct preference optimization: Your language model is secretly a reward model. NeurIPS 2023.
> 2. Zhou, Z., Liu, J., Shao, J., Yue, X., Yang, C., Ouyang, W. and Qiao, Y., 2023. Beyond one-preference-fits-all alignment: Multi-objective direct preference optimization. arXiv preprint arXiv:2310.03708.
>
> > It would be better to include a discussion on the scalability of MPO with respect to computation resources when increasing the number of objectives.
>
> Thanks for the question. From our observation, computational cost increases approximately linearly with the dimensionality of $\lambda$. Specifically, obtaining an approximate optimal $\lambda$ (using 600 iterations of Algorithm 2) requires:
>
> | dim($\lambda$) | A100 GPU hours |
> |:-:|:-:|
> | 2 | 1.8 |
> | 3 | 2.5 |
> | 4 | 3.3 |
>
> This linear scaling indicates that the computational burden remains manageable when increasing the number of objectives.

---

### Decision · Program_Chairs · 2025-05-01

**Decision:**

Accept (poster)

**Comment:**

My recommendation is to accept the paper.

The paper proposes a method for inference-time aggregation of policy models that have been tuned to optimize different preference objectives. The paper shows an equivalence between a max-min RL approach to balancing multiple objectives at tuning time and a post-processing approach that balances single-objective-optimized policies according to a normalized max-min objective at inference time. The authors derive a mirror descent algorithm for learning the parameters of this aggregation, and demonstrate its effectiveness against other multi-objective RL and model averaging alternatives.

Reviewers agreed that the paper addresses an interesting problem, and the solution seems to work. To me, the most serious questions raised were around the novelty of the work, especially wrt [Shi et al 2024], which addressed a similar inference-time alignment problem. However, I found the authors' response to be convincing, specifically that Shi et al do address a different problem formulation, where a specification for linear aggregation is given a priori, whereas this paper addresses max-min aggregation, and addresses the additional technical details (i.e., normalization) necessary for solving the problem in this different context. This being said, I am sympathetic to FxzW's concern about novelty, and hope that the authors make their contributions relative to this prior work clear. In addition to discussion in related work, remarks around the main results in the paper and potentially in the introduction would also be appropriate. It may also be useful to include an example where linear aggregation and max-min aggregation give distinct results.

Beyond these questions of novelty, the methodology seems sound, and with the experiments that were added during the rebuttal phase, adequately tested.

Given that there was some confusion among reviewers about the distinctions between training-time vs inference-time aggregation, I might also suggest that the authors consider how these distinctions could be made clearer in the writing, but this is not essential.